

# A complete representation of uncertainties in layer-counted paleoclimatic archives

Niklas Boers[1,2], Bedartha Goswami[3], and Michael Ghil[1,4]

[1]Geosciences Department and Laboratoire de Météorologie Dynamique (CNRS and IPSL), École Normale Supérieure, and PSL Research University, Paris, France.
[2]Potsdam Institute for Climate Impact Research, Potsdam, Brandenburg, Germany.
[3]Institute of Earth and Environmental Science, University of Potsdam, Potsdam, Germany.
[4]Department of Atmospheric and Oceanic Sciences and Institute of Geophysics and Planetary Physics, University of California, Los Angeles, USA.

*Correspondence to:* Niklas Boers (nboers@lmd.ens.fr)

**Abstract.** Accurate time series representation of paleoclimatic proxy records is challenging because such records involve dating errors in addition to proxy measurement errors. Rigorous attention is rarely given to age uncertainties in paleoclimatic research, although the latter can severely bias the results of proxy record analysis. Here, we introduce a Bayesian approach to represent layer-counted proxy records – such as ice cores, sediments, corals or tree rings – as sequences of probability

distributions on absolute, error-free time axes. The method accounts for both proxy measurement errors and uncertainties arising from layer-counting–based dating of the records. An application to oxygen isotope ratios from the North Greenland Ice Core Project (NGRIP) record reveals that the counting errors, although seemingly small, lead to substantial uncertainties in the final representation of the oxygen isotope ratios. In particular, for the older parts of the NGRIP record, our results show that the total uncertainty originating from dating errors has been seriously underestimated. Our method is next applied to deriving the

overall uncertainties of the Suigetsu radiocarbon calibration curve, which was recently obtained from varved sediment cores at Lake Suigetsu, Japan. This curve provides the only terrestrial radiocarbon calibration for the time interval 12.5–52.8 kyr BP. The uncertainties derived here can be readily employed to obtain complete error estimates for arbitrary radiometrically dated proxy records of this recent part of the last glacial interval.

## 1 Introduction

Time series derived from paleoclimatic proxy records — such as tree rings, ice or marine sediment cores — provide the only available means for quantitative analyses of climate variability on time scales that exceed the last approximately 150 years (Mann and Jones, 2003; Mann et al., 2008). Such records yield indirect information of the past evolution of climatic observables like temperature, carbon concentration or precipitation. This information, however, is provided as a function of the depth in the

record, rather than as a function of time. Depending on the type of proxy record, establishing a precise relationship between



depth and time — i.e., the age–depth model — is a highly nontrivial task (Telford et al., 2004). Widely used techniques include absolute dating methods such as radiometric dating, but also incremental dating methods, such as counting annual layers, also called varves.

In typical age modeling frameworks, the assignment of timestamps to given depths in the proxy core is irregular in time, involves possibly correlated uncertainties, and can in many cases only be performed for a subset of the proxy measurements. The interpolations that become necessary therewith further enhance the final uncertainties. Therefore, in addition to the errors associated with the measurement of a proxy value, there are also substantial uncertainties associated with its dating.

Both types of uncertainties have to be rigorously dealt with, since any conclusions drawn from the resulting time series will strongly depend on the statistical treatment of these uncertainties. For example, apparent abrupt transitions may just be artifacts of the age–depth model construction. Furthermore, particular care needs to be exercised when comparing proxy records obtained from different archives with independent age models. In many situations, e.g. when the accumulation process of the record is taken into account, or for incremental dating techniques, uncertainty distributions at distinct locations of a given record will not be statistically independent.

Surprisingly, out of 93 publications from the year 2008 which involve age–depth models, 65 do not specify if and how uncertainties have been accounted for in the age–depth modeling process (Blaauw, 2010). Scholz et al. (2012) have compared five recent methods to date speleothem records that estimate dating uncertainties also between dated points. For example, such uncertainties can be considered on the basis of Monte Carlo simulations, which fit ensembles of straight lines between the dated points (Scholz and Hoffmann, 2011). Alternatively, one can use a mechanistic-statistical model that combines a deterministic paleoclimate model with a statistical model of the observation process to obtain bifurcation parameters with known error bars (Roques et al., 2014). Furthermore, Breitenbach et al. (2012) have recently proposed a methodology to treat dating uncertainty propagation that is based on repeated interpolation between dated points, and to use the resulting ensemble of age-depth models to translate the uncertainties from the age axis to the proxy axis.

Most recently, Goswami et al. (2014) introduced a Bayesian framework to treat correlated errors in radiometrically determined chronologies. A processed proxy time series resulting from this approach consists of a sequence of probability densities whose domain covers the proxy values, conditioned on prescribed age values. This sequence is represented on an absolute, error-free time axis, a setting that is helpful in many situations. For instance, quantifying rates of change in the time series during prescribed time intervals calls for such an error-free time axis. In addition, an absolute time axis is of the essence when comparing two or more proxy archives with independent chronologies and, in particular, when analyzing the synchronicity of specific events in such archives (Blaauw et al., 2010; Blaauw, 2012).

Existing approaches leading to an error-free time axis are, however, designed for radiometrically dated proxy records (Breitenbach et al., 2012; Goswami et al., 2014). They are thus not directly applicable to proxy records with layer-counting–based chronologies, like ice cores, varved sediments, banded corals, or tree rings. The nature of the chronological uncertainty in such records is fundamentally different from that in radiometrically dated ones. Lotter and Lemcke (1999), for instance, have discussed the statistical problems arising in layer-counted chronologies in the context of annual biochemical varves.



More recently, a model based on discrete random walks was proposed (Rhines and Huybers, 2011) to account for dating errors in the GISP2 ice core proxy record (Alley et al., 1997). Comboul et al. (2014) have also carried out a thorough analysis, based on a probabilistic age model, of dating uncertainties in banded choral archives.

Inspired by the Bayesian approach for radiocarbon-dated archives of Goswami et al. (2014), we introduce here a similar approach that is specifically designed to account for the uncertainties arising in layer-counted chronologies. Our approach also relies on a Bayesian framework to propagate all uncertainties to the proxy axis, and represents the proxy record as a sequence of probability densities on a prescribed, error-free time axis.

The key observation of our approach is that the probability distribution of a proxy value $x$, given a calendar age $t$, can be expressed in terms of the probability distributions of $x$ and $t$, conditioned on the measured depth in the proxy archive $z$:

$$p(x|t) = \frac{\int p(x|z)p(t|z)dz}{\int p(t|z)dz} \tag{1}$$

This equation reveals that $p(x|t)$ is in fact the normalized average of $p(x|z)$ over all depths $z$, weighted by the respective contributions $p(t|z)$, which are considered as functions of the proxy depth $z$. The details appear in the Methods section and are illustrated in Fig. 1A.

Ideally, precise estimates of the measurement uncertainty distribution $p(x|z)$ and the dating uncertainty distribution $p(t|z)$ would be reported together with the proxy data themselves. In practice, however, this is rarely the case, and typically both distributions are assumed to be Gaussian, in which case it would suffice to report the mean values and standard deviations for the proxy and age measurements, respectively. We will show below how the specific functional form of the uncertainty distributions strongly impacts the time series representation of the corresponding proxy record.

In order to emphasize the need for a rigorous handling of dating uncertainties in layer-counted proxy records, we first test and illustrate our method by applying it to $\delta^{18}O$ isotope ratios obtained from the North Greenland ice core (NGRIP) project. For this record, a layer-counted chronology exists for the past 60 kyr (Svensson et al., 2008; Rasmussen et al., 2014). Our results will reveal that the overall uncertainties due to dating errors in this record have been substantially underestimated.

Thereafter, we further generalize our approach in order to represent the increments $\Delta x$ of a proxy record $x$ between distinct time steps; doing so is important, for instance, when empirically estimating stochastic differential equation models from such records (Ditlevsen et al., 2005; Kwasniok, 2013; Krumscheid et al., 2015; Mitsui and Crucifix, 2017; Boers et al., 2017). In layer-counted chronologies, uncertainties accumulate toward the more remote past because one typically starts counting at the top of the core, i.e. at the most recent layer. Furthermore, the identification of periodic layers, such as seasonal varves, will become increasingly more uncertain due to accumulation processes and the typically decreasing quality of the record with increasing depth. This increasing uncertainty may lead to very large absolute uncertainties for the dating of the archive's remote past. When analyzing relative changes $\Delta x$, however, only the relative dating uncertainties matter (cf. Methods).

Third, we apply our method to $\Delta^{14}C$ measurements obtained from the sediments of Lake Suigetsu, Japan, which allow for varve counting for the time interval $10.2 - 40.0$ kyr BP (Marshall et al., 2012; Schlolaut et al., 2012; Staff et al., 2013). The $\Delta^{14}C$ record from Lake Suigetsu is used to calibrate measured $^{14}C$ ages with respect to the varve-counted chronology,



and provides the only available fully terrestrial calibration curve for radiometric dating of proxy records prior to 12.5 kyr BP (Bronk Ramsey et al., 2012).

Using our time-to-proxy method, we derive the overall uncertainties for this extensive calibration curve, taking into account both $^{14}$C measurement errors and errors stemming from the varve-counted Suigetsu chronology. Finally, it is shown how the

overall uncertainties of the Suigetsu calibration curve can be used to obtain complete error estimates for arbitrary radiometrically dated proxy records that cover the interval from 40.0 kyr BP to the present.

## 2  Data

### 2.1  NGRIP ice core data

We employ a proxy record of $\delta^{18}$O ratios from the North Greenland Ice Core Project (NGRIP). The $\delta^{18}$O ratios obtained from

ice cores are commonly interpreted as proxies for surface air temperature variability (Johnsen et al., 1992; Dansgaard et al., 1993; Johnsen et al., 2001; Andersen et al., 2004). The layer-counted chronology of this record is the Greenland Ice Core Chronology 2005 (GICC05) (Svensson et al., 2008), which starts at 60 kyr before AD 2000, abbreviated as b2k herein. We use the published version of this record, with measurement values of $\delta^{18}$O ratios reported at a temporal resolution of 20 yr. Dating uncertainties in terms of maximum counting errors (MCE) are given for each of these time steps in the published dataset

(Andersen et al., 2006; Rasmussen et al., 2006). As a result of the layer-counted dating, the MCE increase monotonically toward the past; see (Svensson et al., 2008; Rasmussen et al., 2014) and Fig. 1 herein.

### 2.2  Suigetsu lake sediment data

Sediment cores obtained from Lake Suigetsu, Japan, allow for a floating, varve-counted chronology of the approximate interval $10.2 - 40.0$ kyr BP, where BP refers to the year 1950 (Marshall et al., 2012; Schlolaut et al., 2012; Staff et al., 2013). Counting

errors are reported in terms of $1\sigma$, from which we obtain the corresponding maximum counting error as MCE $= 2\sigma$ (Andersen et al., 2006). Using this chronology and combining it with speleothem data for its more recent past, a comprehensive $^{14}$C record was obtained that provides a unique calibration curve for atmospheric radiocarbon age measurements over the time interval $10.2 - 52.8$ kyr BP (Bronk Ramsey et al., 2012).

We used the $\Delta^{14}$C data, as well as the $^{14}$C age measurements, together with the corresponding varve-counted time stamps

from Bronk Ramsey et al. (2012). We did not consider the interval $40.0 - 52.8$ kyr BP, also included in (Bronk Ramsey et al., 2012), because the chronology for this segment is not directly varve-counted, but extrapolated from the varve chronology of the more recent time interval $10.2 - 40.0$ kyr BP.

## 3  Methods

In general, a proxy record consists of a set of depths $z_i$ in the archive, as well as proxy values $x_i$ and calendar ages $t_i$, which are

both measured at depths $z_i$. In the case of interest for this study, the measurement of the calendar age is performed by counting





layers that are a priori assumed to correspond to some known periodicity, such as annual layers in ice cores, sediment layers in lakes or growth rings in trees. It is already evident at this point that the uncertainties associated with the counting process are monotonically increasing in reverse time, as well as being strongly correlated.

The quantity we are interested in is the conditional probability distribution $p(x|t)$ of the proxy values $x$, given prescribed, error-free calendar ages $t$. The key observation is that this can be written in terms of $p(x|z)$, the probability distribution accounting for measurement uncertainties of the proxy value $x$ at a given depth $z$, together with $p(t|z)$, the probability distribution of a calendar age $t$ given a depth $z$, which accounts for the dating uncertainties (Goswami et al., 2014):

$$p(x|t) = \int p(x|z)p(z|t)dz = \int p(x|z)p(t|z)\frac{p(z)}{p(t)}dz \ . \tag{2}$$

The first equality above is due to the chain rule for probabilities, and the second is due to Bayes' Theorem. Assuming so-called flat priors, i.e. a uniform distribution for the depths $z$, the fraction $p(z)/p(t)$ is merely a normalization constant; it can be determined by observing that, by definition, $\int p(x|t)dx = 1$ and $\int p(x|z)dx = 1$. We thus arrive at

$$p(x|t) = \frac{\int p(x|z)p(t|z)dz}{\int p(t|z)dz} \ . \tag{3}$$

In practical applications, only a finite number $N$ of observations will be available. In such cases, the above expression will be approximated by a corresponding Riemann sum over the depths $z_i$ at which measurements have been performed:

$$p(x|t) = \frac{\sum_{i=1}^{N} r_j p(x|z_i)p(t|z_i)}{\sum_{i=1}^{N} r_i p(t|z_i)} \ , \tag{4}$$

where $r_1 = z_2 - z_1$, $r_N = z_N - z_{N-1}$, and $r_i = (z_{i+1} - z_{i-1})/2$ for $1 < i < N$.

For a prescribed calendar age $t$, the probability of a specific proxy value $x$ thus involves the uncertainty distributions of $x$, as well as the uncertainty distributions of $t$, at all depths $z_i$ (cf. Fig. 1A). Note that the probability distributions for the proxy values are derived as marginal distributions given prescribed calendar ages; the latter can thus be freely chose, e.g. equidistantly with a desired temporal resolution.

In addition to $p(x|t)$, we are interested in $p(\Delta x|t,t')$, where $\Delta x$ denotes the change of the proxy value $x$ between time $t$ and a later time $t'$, i.e. $\Delta x = x_{t'} - x_t$. These could be adjacent time steps, in which case $\Delta x$ would indicate the change of the proxy value $x$ per (arbitrary) unit of time. Precise estimates of the latter are crucial, for instance, in data-driven modeling of the temporal evolution of the proxy value $x$ in terms of differential equations, either deterministic or stochastic (Ditlevsen et al., 2005; Kwasniok, 2013; Krumscheid et al., 2015; Mitsui and Crucifix, 2017; Boers et al., 2017).

As in Eq. (2) above, $p(\Delta x|t,t')$ can be expanded as

$$p(\Delta x|t,t') = \int \int p(\Delta x|z,z')p(t,t'|z,z')dzdz' \ . \tag{5}$$

For arbitrary random variables $a$, $b$, $c$, and $d$, Bayes' Theorem implies that

$$p(a|b,c,d) = \frac{p(a,b,c,d)}{p(b,c,d)} = \frac{p(c,d|a,b)p(a,b)}{p(b,c,d)}$$





and therefore

$$p(t'|t,z,z') = \frac{p(z,z'|t,t')p(t,t')}{p(t,z,z')} \ .$$
(6)

Inserting this into Eq. (5) and assuming a uniform prior $p(t,z,z')$, we obtain

$$p(\Delta x|t,t') = \frac{\int\int p(\Delta x|z,z')p(t'|z,z',t)dzdz'}{\int\int p(t'|z,z',t)dzdz'} \ ;$$
(7)

here the same reasoning as for Eq. (3) applies concerning the normalization.

## 4    Results

### 4.1    Dating uncertainties in the NGRIP $\delta^{18}$O record

The details of the NGRIP proxy record's layer-counted chronology and associated uncertainties have already been discussed at considerable length (Andersen et al., 2006; Rasmussen et al., 2006; Svensson et al., 2008). The further one goes into the past,

the more the layer-counting errors accumulate, leading to $\text{MCE} = 2601$ yr at a layer-counted age of 60 kyr b2k (Fig. 1B). As explained in detail below, the precise dating uncertainty distribution $p(t|z)$ is typically unknown.

Given the number of uncertain layers $L = 2 \cdot \text{MCE}$, each counted as $1/2 \pm 1/2$ yr, uncorrelated counting errors would lead to a simple error estimate in terms of a normal distribution with $\sigma = \left(L(1/2 \ \text{yr})^2\right)^{1/2} = \left(2 \cdot \text{MCE}\right)^{1/2}/2$. This naive estimate is, however, very unrealistic since the errors are much more likely to be correlated (Andersen et al., 2006). Several authors have

suggested assuming Gaussian functional forms for $p(t|z)$, with standard deviations set to $\sigma = \text{MCE}/2$ (Andersen et al., 2006; Rasmussen et al., 2006; Svensson et al., 2008).

This assumption accounts, to some degree, for correlated errors and it gives quite conservative error estimates, but it also postulates a very specific, Gaussian form for the uncertainty distribution. In an even more conservative vein, one has to admit that nothing is known about the way the counting errors associated with uncertain layers depend on each other. This assumption

would lead to a uniform distribution of dating uncertainties, namely:

$$p(t|z_i) \sim \mathcal{U}(t(z_i) - \text{MCE}, t(z_i) + \text{MCE})$$
(8)

We apply our methodology to represent the NGRIP $\delta^{18}$O record (Fig. 2A) as a sequence of probability densities on an error-free time axis with equidistant 20-year steps. Since the proxy measurement uncertainties are not reported in the published version of this dataset, we assume for $p(x|z)$ a normal distribution, centered at the reported values and with very small $\sigma =$

0.01 per mille. The results we present in the following would not change if the proxy measurement errors were completely neglected by setting $p(x|z_i) = \delta(x - x(z_i))$, the Dirac delta distribution located at the measured value $x(z_i)$.

We employ three different functional forms for the age uncertainty distribution $p(t|z)$ in order to illustrate how this choice impacts the final representation. First, the rather unrealistic choice of a normal distribution with $\sigma = \left(2 \cdot \text{MCE}\right)^{1/2}/2$, which





corresponds to independent dating errors, is used in Fig. 2B. In this case, the variability of the $\delta^{18}$O record — as seen in the blue line of Fig. 2A, which does not account at all for dating uncertainties — is retained by $\hat{x}(t)$, the expectation value of $\delta^{18}$O with respect to $p(x|t)$ (red solid line in panels B–D of the figure). Due to the small $\sigma$ imposed on $p(t|z)$ when assuming uncorrelated errors, the densities $p(x|t)$ are very localized, and hence the blue shading appears merely as single points in Fig. 2B.

In contrast, as seen in Fig. 2C, using a normal distribution with $\sigma = \text{MCE}/2$ for the counting error distribution $p(t|z)$, as proposed in (Andersen et al., 2006; Rasmussen et al., 2006; Svensson et al., 2008), leads to substantially larger overall uncertainties in $p(x|t)$. Moreover, this uncertainty grows significantly into the past, due to the accumulation of errors in the layer-counting process.

Finally, using a uniform distribution $p(t|z)$ (cf. Eq. (8) and Fig. 2D) yields an uncertainty range similar to the one in Fig. 2C.
The short-term variability of $\hat{x}(t)$ that is left in these two panels, however, differs substantially between the representations using a normal and a uniform distribution, respectively.

The assumption of uncorrelated dating errors in the layer-counted chronology GICC05 of the NGRIP records is hardly justifiable (Andersen et al., 2006; Rasmussen et al., 2006), and therefore statistically dependent dating errors should be assumed when choosing the age uncertainty distribution $p(t|z)$. The MCE of the layer-counted chronology of the NGRIP record reaches
2601 yr at 60 kyr b2k; it is hence smaller than 4%, and confirms therewith the high accuracy of the dating process by counting annual layers. Still, although the MCE seems small, it leads to strong growth in the overall uncertainties in the final representation of the $\delta^{18}$O record. For the older parts of the record, these uncertainties affect more and more a statistically sound estimation of proxy values; see Figs. 2C and 2D. This difficulty implies that great care has to be exercised when comparing and, in particular, trying to align specific events among two or more distinct proxy records.

Comparing the expected values $\hat{x}(t)$ of $\delta^{18}$O in Figs. 2C and 2D — obtained using a normal and a uniform distribution for $p(t|z)$, respectively — shows similar long-term variability, but significant differences in the short-term variability of their temporal evolution: the use of a normal dating error distribution leads to a much smoother temporal evolution of $\hat{x}$ than a uniform distribution. This apparent smoothness is, however, an artifact of employing a normal distribution for $p(t|z)$.

The latter functional form is not justified by the dating process itself, since the actual dependence of the dating errors is
unknown. We hence argue for the use of a uniform distribution for $p(t|z)$; first, because it provides a more accurate representation of the dating uncertainties when their dependencies are unknown, and second, because it avoids at the same time the probably spurious smoothness caused by using a normal form of $p(t|z)$. In practice, the true distribution of dating uncertainties is impossible to derive for chronologies based on counting periodic layers. Most likely, it will be a mixture of two extremes: on the one hand a normal distribution, which corresponds to weak correlations between counting errors, and on the other hand
a uniform distribution, which corresponds to maximum correlations between these errors. In principle, one could use a convex linear combination of the two proposed forms, $p(t|z) = \alpha\mathcal{N} + (1-\alpha)\mathcal{U}$, properly normalized, or some other probability mixture, to obtain a desired degree of smoothness of the evolution of $\hat{x}(t)$. Of course, any a priori knowledge of uncertainties that can be derived from the dating process should be included here.

The distribution $p(x|t)$ is typically multimodal for given $t$ and $p(t|z)$. If so, the expectation value $\hat{x}$ itself can have rather
low probability, cf. inset of Fig. 2D. To avoid misguided interpretations, the proxy record should thus be visualized in terms of



$p(x|t)$ (blue shading) and not in terms of $\hat{x}$ (red solid lines). The latter is only shown here to enable a direct comparison with the traditional representation of the record as a unique, scalar time series, cf. Fig. 2A.

If one is not interested in a representation of the proxy record itself, but rather in the representation of its relative changes, only the relative counting errors matter. In this case, our method should be applied to the one-step increments of the NGRIP

$\delta^{18}$O record, i.e. $\Delta x_i = x_{i+1} - x_i$; see Methods section. Because the relative counting errors do not accumulate over time, a very precise estimation of these *relative* changes is in fact possible, cf. Fig. 3.

The relative representation in Fig. 3B of the increments $\Delta x_i = x_{i+1} - x_i$ of the $\delta^{18}$O record, using the relative counting errors between subsequent measurement points, underscores the fact that the large overall uncertainties observed for the absolute representations in Figs. 2(C,D) are due to the accumulation of dating errors. The finding that a precise estimation of the

increments is possible indicates that the short-term variability of the $\delta^{18}$O in terms of relative changes can be estimated with a high degree of certainty. This result explains the success of attempts to derive dynamical models from paleoclimatic records (Ditlevsen et al., 2005; Kwasniok, 2013; Krumscheid et al., 2015; Mitsui and Crucifix, 2017; Boers et al., 2017) despite neglecting the dating uncertainties.

Nevertheless, the short-term variability is strongly smoothed in the absolute representation of the record (Figs. 2C and 2D);

such smoothing correctly reflects the fact that absolute dating of the more remote parts of the record is jeopardized by accumulating counting errors. An absolute representation should be used, for instance, when comparing and aligning several proxy records on an absolute time scale, while a relative representation of increments should be used when one focuses on the high-frequency variability and relative changes of the proxy record under consideration.

## 4.2   Dating uncertainties in the Suigetsu $\Delta^{14}$C calibration curve

We restrict ourselves to the time interval $10.2 - 40.0$ kyr BP, for which a varve-counted chronology of the Lake Suigetsu sediment record exists (Marshall et al., 2012; Schlolaut et al., 2012; Staff et al., 2013). As for the NGRIP record, errors associated with the varve-counting accumulate toward the past: the largest error, for the varve counted part, is reported to be $\sigma = 1707$ yr at a varve-counted age of $38964$ yr; see also Fig. 1C. Note that, in accordance with the discussion of the NGRIP record and Fig. 2, we set MCE $= 2\sigma$. For the Suigetsu record, we choose a uniform time axis with $50$ yr increments.

The $\Delta^{14}$C series of the Suigetsu sediments, shown here in Fig. 4A, has been used as a comparison curve to fit radiometrically dated archives — such as speleothem data (Hoffmann et al., 2010; Southon et al., 2012) — onto the Suigetsu varve chronology. As done in the previous section for NGRIP, we represent in Fig. 4B the Suigetsu $\Delta^{14}$C record on an error-free time axis, accounting for the reported $\Delta^{14}$C measurement errors, as well as for the reported counting errors of the varve chronology. Due to the accumulating counting errors, the overall uncertainties in the $\Delta^{14}$C values derived in this way become considerably

larger the further one proceeds into the past.

The radiocarbon age curve in Fig. 4C, considered as a function of the varve-counted ages, is itself a paleoclimatic proxy. We can therefore use our method to derive a representation of the overall uncertainties in this radiocarbon age calibration curve, originating from both radiocarbon age measurement errors and varve counting errors. This representation yields the uncertainty



distribution $p(t_{\mathrm{rc}}|t)$ of the radiocarbon age $t_{\mathrm{rc}}$, given the true calendar age $t$, as plotted in Fig. 4D. The largest uncertainty of the radiocarbon age, quantified as the interquartile range, is $3487$ yr, observed at a calendar age $t = 34\,227$ yr.

Our representation of the Suigetsu $\Delta^{14}\mathrm{C}$ values in Fig. 4B reveals that, despite the seemingly small varve counting errors, of less than $5\%$, considerable uncertainties are involved in the radiocarbon age calibration for which this record is used as a

standard comparison curve. These uncertainties propagate to the final radiocarbon age model in Fig. 4D, which is proposed to be used to calibrate ages in other, radiocarbon-dated proxy archives.

The uncertainty distribution $p(t_{\mathrm{rc}}|t)$ of the radiocarbon ages, as derived from our method, can be directly used to obtain complete uncertainty estimates when analyzing arbitrary radiometrically dated proxy archives: Measuring a proxy variable $v$ in a given radiocarbon-dated archive, the probability distribution of $v$, given a true calendar age $t$, can be expanded in terms of

the radiocarbon age $t_{\mathrm{rc}}$ as

$$p(v|t) = \int p(v|t_{\mathrm{rc}})p(t_{\mathrm{rc}}|t)dt_{\mathrm{rc}} . \tag{9}$$

Here, $p(v|t_{\mathrm{rc}})$ is the distribution of proxy values $v$, measured at radiocarbon ages $t_{\mathrm{rc}}$ in the other archive, and $p(t_{\mathrm{rc}}|t)$ is the distribution of total uncertainties in the Suigetsu radiocarbon age model, shown in Fig. 4D. In order to facilitate the incorporation of these uncertainties in the estimation of a proxy variable from any radiometrically dated archive, we provide

the values of $p(t_{\mathrm{rc}}|t)$ for different temporal resolutions in the online supplementary material.

In addition, one may be interested in the probability of a true calendar age $t$, given a measured radiocarbon age $t_{\mathrm{rc}}$. For arbitrary radiocarbon-dated proxy archives, this can be obtained from the uncertainty distribution $p(t_{\mathrm{rc}}|t)$ of the Suigetsu radiocarbon ages, via Bayes' Theorem:

$$p(t|t_{\mathrm{rc}}) = \frac{p(t_{\mathrm{rc}}|t)p(t)}{p(t_{\mathrm{rc}})}$$

Here, $p(t)$ can be assumed to be an uninformative prior, while $p(t_{\mathrm{rc}})$ reflects the uncertainty distribution of the measurement of the radiocarbon age $t_{\mathrm{rc}}$.

## 5   Conclusions

We have introduced statistically rigorous representations of layer-counted proxy records as sequences of probability distri-

butions on error-free time axes. In such records, the calendar age is determined by counting layers, which are assumed to correspond to a known periodicity, such as annual layers in ice cores, varved sediment layers, banded corals, or growth rings in trees. Our approach, which is rooted in Bayesian statistics, takes into account the uncertainties in both proxy measurement and dating. Such an unambiguous representation of uncertainties is crucial, for instance, when comparing proxy records obtained from different archives. Our results indicate that the effects of dating uncertainties in paleoclimatic archives have been strongly

underestimated, and emphasize the urgent need for an adequate statistical representation of proxy records with immanent dating uncertainties in order to prevent misleading interpretations: The common representation of such proxy records as time series



on the mean or median values of the age distributions induces a strong bias, suggesting a much higher degree of accuracy than is actually warranted.

First, we illustrated our method by applying it to the $\delta^{18}$O record obtained from the NGRIP ice core. A representation of this record on an absolute time scale (Fig. 2) revealed that the statistically dependent, cumulative errors associated with the

layer-counted dating, although seemingly small, lead to growing overall uncertainties the further one digs into the past.

This finding calls for great caution when aligning proxy records from distinct sources on a common, absolute time scale. On the other hand, a representation of the increments of the $\delta^{18}$O record between subsequent time steps (Fig. 3) is possible with a comparably high degree of certainty, because in this case only the relative counting errors from one measurement point to the next are relevant. These increments and their representation are of interest if one focuses on the short-term variability.

In particular, modeling the associated time series in terms of differential equations, whether deterministic or stochastic, is relatively insensitive to such errors.

Second, we applied our method to the radiocarbon calibration curve recently derived from annually layered sediment cores of Lake Suigetsu, Japan. To date, this is the only available terrestrial radiocarbon record that can be used to calibrate other radiometrically dated proxy archives for ages prior to 12.5 kyr BP.

The proposed method allowed us to propagate the uncertainties associated with the layer-counted Suigetsu chronology, as well as the ones associated with the radiocarbon measurements, to the final radiocarbon calibration curve. The resulting sequence of probability distributions represents the overall uncertainties of the calibration curve. This sequence can and should be used to obtain accurate uncertainty estimations in arbitrary radiocarbon-dated proxy archives for which the Suigetsu radiocarbon calibration curve is employed. Furthermore, our approach allows one to compute the uncertainty distribution of the true

calendar age, given a radiocarbon age estimate.

*Data availability.* The ice core data used in this study are publicly available at http://www.iceandclimate.nbi.ku.dk/data/. The Suigetsu sediment data can be downloaded from http://www.suigetsu.org/publications.html. The uncertainty distribution of the radiocarbon ages derived from the Suigetsu sediment data will be published online once this paper is published.

*Author contributions.* NB conceived the study, carried out the analysis, and prepared the manuscript. All authors discussed the results, draw

conclusions, and contributed to editing the manuscript.

*Competing interests.* The authors declare that they have no competing financial interests.





*Acknowledgements.* NB acknowledges funding by the Alexander von Humboldt Foundation and the German Federal Ministry for Education and Research. MG was partially supported by ONR's Multidisciplinary Research Initiative (MURI) grant N00014-16-1-2073 and by NSF grant OCE-1243175.



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



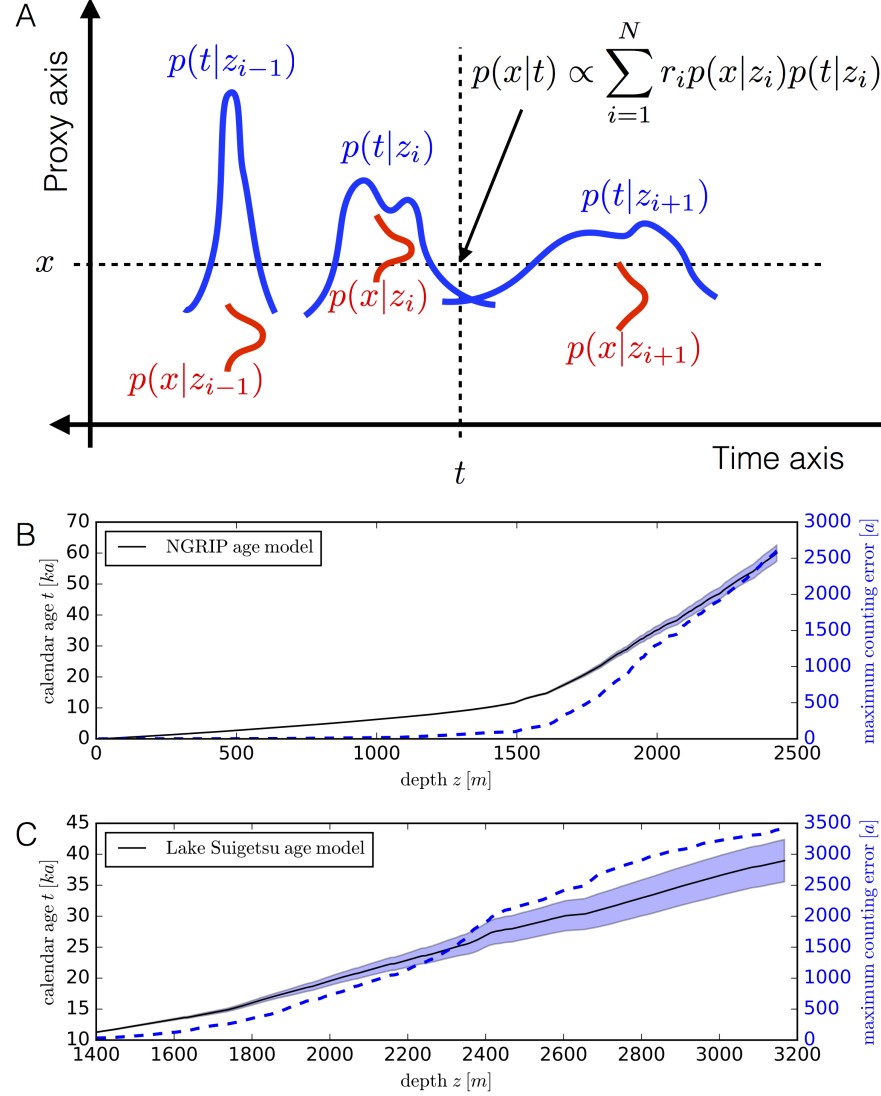

**Figure 1. Sketch of converting time axis uncertainties into proxy axis ones, and NGRIP and Suigetsu age models.** A. Scheme illustrating the key ideas of the proposed approach to represent proxy records dated by counting layers as sequences of probability densities. In this sketch, the measurements of the proxy $x$ and the corresponding calendar age $t$ are indicated at three different depths $z_{i-1}$, $z_i$, and $z_{i+1}$. The uncertainty distributions $p(x|z)$ of the proxy measurement processes are depicted in red, and those of the calendar ages $p(t|z)$ in blue. For a prescribed time $t$, the final probability distribution $p(x|t)$ of proxy values is given as the normalized average of $p(x|z)$, weighted by the corresponding contributions of all age uncertainty distributions $p(z|t)$; see the Methods section for further details. B. The age–depth relation of GICC05 for the NGRIP ice core (solid black line) (Andersen et al., 2006; Rasmussen et al., 2006). The associated uncertainties are quantified as the maximum counting error MCE, given in years corresponding to annual layers; they are depicted as blue shading around the age–depth relation, and additionally as the blue dashed line with scale shown on the right-hand side. In this chronology, uncertain layers are counted as $1/2 \pm 1/2$ yr (Andersen et al., 2006; Rasmussen et al., 2006), which implies that the MCE is half the number of uncertain layers. C. The age–depth relation of the varve-counted segment of the Suigetsu lake deposit record. Note how the uncertainties increase monotonically for both records, because the counting process causes them to accumulate toward the past.





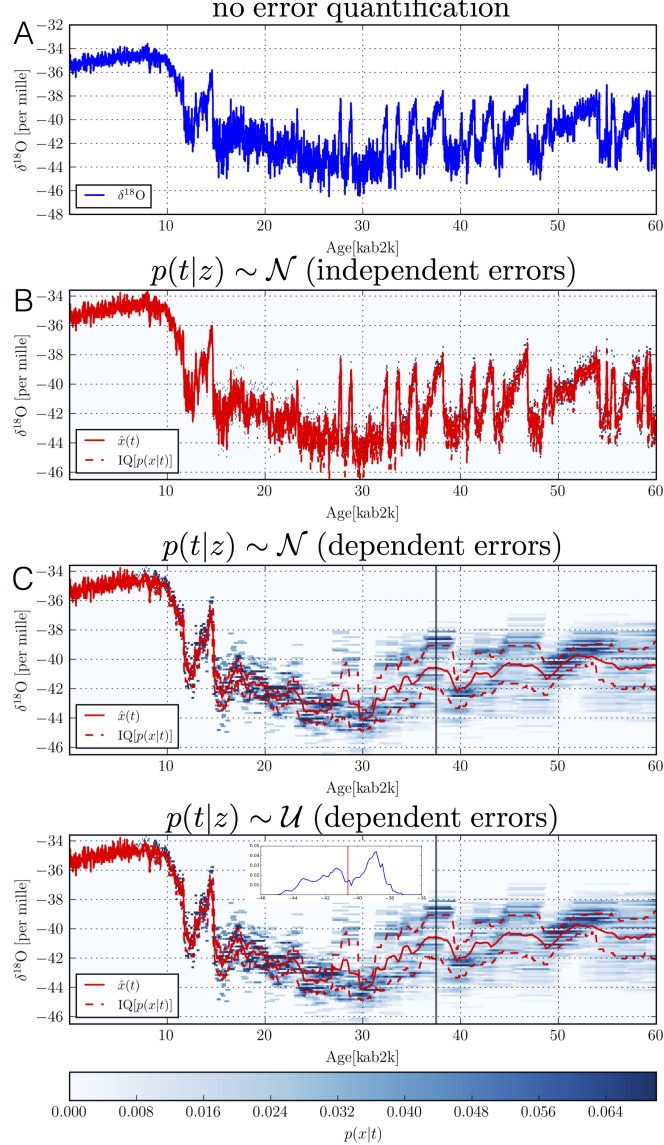

**Figure 2. Representation of the NGRIP $\delta^{18}$O record, taking into account absolute counting errors in the dating.** A. Traditional representation of the record with no accounting for dating uncertainties at all: the time steps are obtained as the mean values of the corresponding dating uncertainty distributions. B. Representation of the same dataset as a sequence of probability densities, derived via Eq. (4); here we used a normal distribution with $\sigma = \left(2 \cdot \text{MCE}\right)^{1/2}/2$, which corresponds to uncorrelated errors, for the age uncertainty distribution $p(t|z)$. The blue shading showing the distributions $p(x|t)$ as a function of the prescribed time $t$ is, due to the small $\sigma$ for uncorrelated errors, only visible as single blue points. In this case, the time-dependent expectation value of $x = \delta^{18}$O with respect to $p(x|t)$, denoted by $\hat{x}(t)$ (red), closely resembles the traditional representation in panel A. C. Same as panel (B), but using a normal distribution with $\sigma = \text{MCE}/2$ to represent the counting errors. D. Same as (B), but using a uniform distribution of width $2 \cdot \text{MCE}$ to represent the counting errors. The inset shows $p(x|t)$ (blue) and $\hat{x}(t)$ (red) at $t = 37.5$ kyr b2k, also indicated by the grey vertical line in the main panel. Note that, in panels C and D, the spread of the distributions $p(x|t)$ — as quantified by the interquartile (IQ) range of $p(x|t)$ (dashed red line) — becomes wider the further one goes into the past. This increase in the spread of $p(x|t)$ is also reflected by a decrease of the high-frequency variability of its expectation values $\hat{x}(t)$.





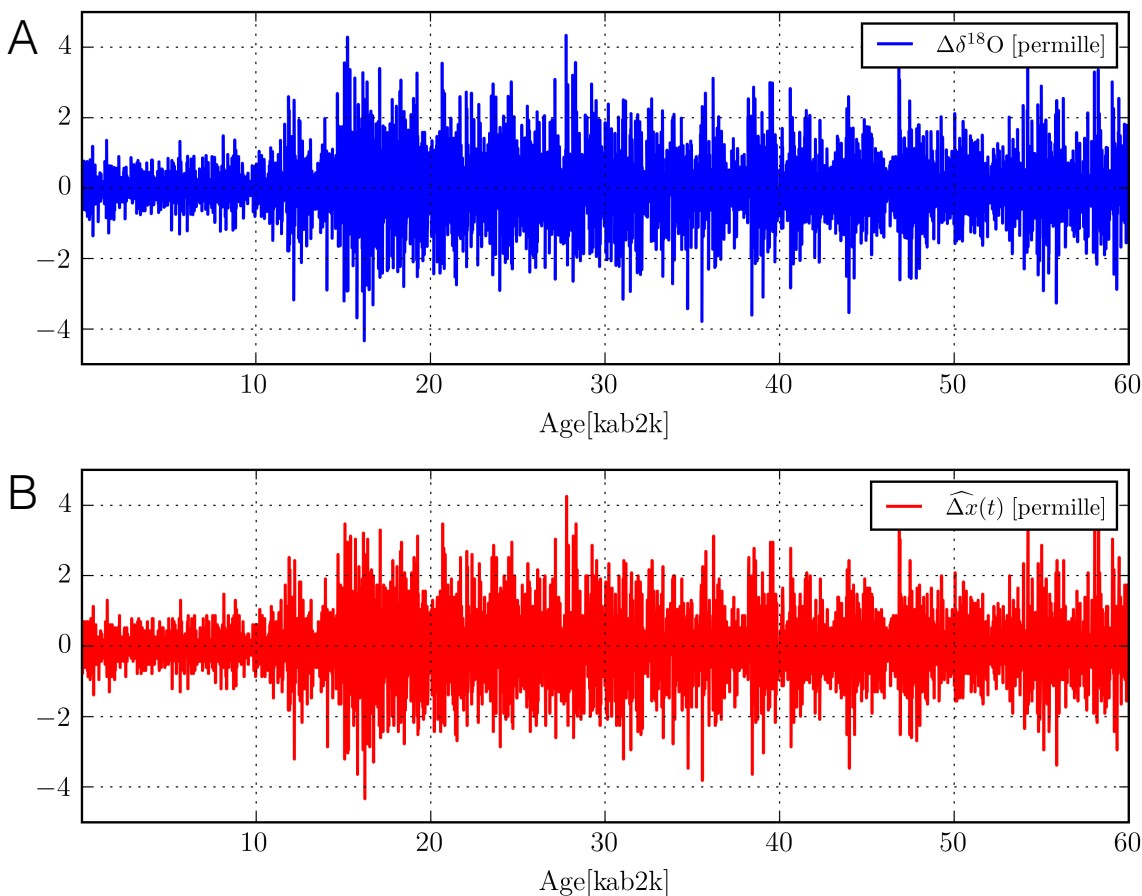

**Figure 3. Representation of the one-step increments $\Delta\delta^{18}O$ of the NGRIP record, taking into account the relative counting errors.** A. Traditional representation of the increments $\Delta\delta^{18}O$ between subsequent time steps; the time steps are obtained as in Fig. 2. B. Representation of the $\delta^{18}O$ increments between subsequent time steps as a sequence of probability densities, derived via Eq. (7). Note that the relative counting errors are sufficiently small to obtain very sharp, unimodal distributions $p(\Delta x | t, t+1)$. For this reason, only the expectation value $\widehat{\Delta x}(t)$ (red line) of the increments with respect to $p(\Delta x | t, t+1)$ is shown in this case.



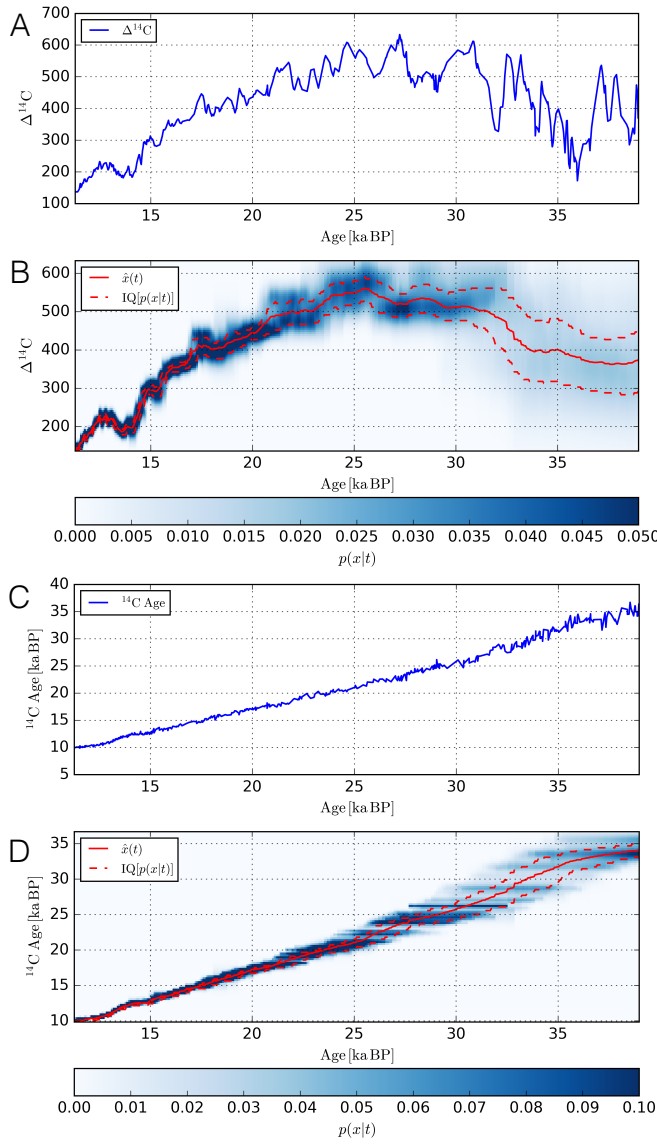

**Figure 4. Representation of the Suigetsu lake sediment data, taking into account proxy uncertainties as well as absolute counting errors in the dating.** A. Traditional representation of the $\Delta^{14}$C time series, without accounting for uncertainties. The time span corresponds to the interval for which a varve-counted chronology is available. B. Representation of the $\Delta^{14}$C from the Suigetsu lake sediment dataset as a sequence of probability densities, derived via Eq. (4). As for the NGRIP record (Fig. 2), a uniform distribution of width 2MCE is used for the age uncertainty distribution $p(t|z)$. C. The radiocarbon ($^{14}$C) calibration curve derived from the Suigetsu record, without accounting for uncertainties. D. Representation of the Suigetsu radiocarbon calibration curve as sequence of probability densities, taking into account radiocarbon measurement errors as well as errors originating from varve counting.