# Peer review of "A complete representation of uncertainties in layer-counted paleoclimatic archives"

_Climate of the Past, 2017_

## Author Comment (AC1) · 26 May 2017

Unfortunately, due to a typo, I uploaded a wrong Fig. 2C in the original manuscript submission; the correct Fig. 2 is attached to this comment. The caption in the main text is still correct. I apologize for the confusion this might have caused.

Niklas Boers
* * *
[Figure]

[Figure]

Fig. 1. Corrected Figure 2

[Figure]

---

## Referee Comment (RC1) · Anonymous Referee #1 · 22 Jun 2017

In this paper a probabilistic representation of uncertainties in the dating of layer-counted paleoclimatic records is presented. The model is based on a Bayesian approach to obtain the probability of a given value of the proxy $x$ at the past time $t$, expressed as a conditional probability $p(x|t)$. This is an unusual representation, since the dating problem is usually expressed in terms of the uncertainty in time (date) $t$ as a function of the depth in the record $z$; $p(t|z)$. By Bayes theorem these are connected, and one is obtained from the other by the standard technique of ignoring the prior distributions (flat priors), and by invoking the conditional probability density of the proxy conditioned on the depth $p(x|z)$. From the mathematical point of view this is all formally correct and consistent, but from a physical point of view it is surprising that $p(x|z)$, which represents the measurement error on the proxy $x$ in the record at depth

$z$, should contain much information on the dating uncertainty for the depth $z$. Anyway, when presenting results the relevant probability $p(t|z)$ is indeed in focus. The main results are presented in Figure 2 (NGRIP) and Figure 4 (Suigetsu lake), where $p(x|z)$ actually does not play a role.

I will focus on the ice core record, which I am most familiar with. Here the dating uncertainty is estimated by (Andersen et al., 2006) by reporting the maximum counting errors (MCE) in the annual layer counting $(t_i = i)$, as a function of depth $z_i$. Three models for the probability density $p(t|z_i)$ are applied: Normal distribution $\mathcal{N}(i, \sqrt{MCE(i)})$ for uncorrelated counting errors, $\mathcal{N}(i, MCE(i))$ for correlated counting errors and $\mathcal{U}(i - MCE(i), i + MCE(i))$ for strongly correlated errors. The results for the latter two are very similar, which is reassuring, since it is difficult to argue for one over the other.

I have two main concerns: Firstly, the results are presented in terms of the record $\hat{x}(t) = \int x p(x|t) dx$. This curve has very little to do with the real proxy record. Especially as the increasing uncertainty with age smears the record. In this particular record with jumps between two states at a time scale of a few thousand years comparable to the dating uncertainty, the statistical mean becomes meaningless. This is also noted by the authors. The problem is reflected in the (very) small insert in Figure 2D, where at a certain time the probability density becomes bimodal. Secondly, the usefulness of the representation in terms of the blue lines representing the dating uncertainty in Figures 2C and D should also be explained. To me it seem that just two figures with time scales corresponding to the upper and lower edges of the blue MCE bands in Figure 1 B .

I can recommend publication.

Minor points:

A discussion on a possible skewness in the distribution $p(t|z)$ could be added: One should expect that the probability of missing an annual "peak" (local maximum) it the curve (say, due to diffusion, or voids in the record.) is different from the probability in

detecting an extra peak or shoulder in the record not representing an annual maximum. (say, due to an anomalous seasonal cycle).

P2L2: The phrase "Varves" is usually only associated with rock or clay sedimentary annual layers, not tree rings or ice cores.

P2L9: The dating uncertainty does not by itself do much to create artificial abruptness in transitions. Such artifacts are more related to disturbances in the record (voids, foldings etc.)

P5L16: The factor $\frac{1}{2}$ in the last formula beats me. Consider using $dz_i$ og $\Delta z_i$ rather than $r_i$, also in formula in Figure 1. It would help readability.

P5L19: chose $\rightarrow$ chosen

P7L10: It is very difficult to see the substantial differences between Fig 2, C and D.

Figure 1 A: I propose to make a 3D figure (see my sketch).

Figure 1 B: Explain the change in slope at 1500 m (transition from Holocene to glacial ice).

Figure 2: Use same scale (y-axis) in A as in B-D. (same for Fig 4 A and B).
* * *
**Fig. 1.**

---

## Referee Comment (RC2) · Anonymous Referee #2 · 23 Jun 2017

This is an interesting contribution to enhancing our understanding of the chronological uncertainties of layer-counted fossil archives. Several sections need more clarification or expansion, and the interpretation and implications of the $\delta^{14}$C record need to be checked. Pending these changes I would recommend publication.

The potential for asymmetric errors in layer counting needs to be discussed. There is a probability of missing annual layers, and a different probability of falsely assuming layers are annual. How was this approached in NGRIP and Suigetsu, and how might this affect the errors and your analysis?

p8 lines 15–16, it could be mentioned that *if* for example a tephra were to be found in the Greenland ice cores or Suigetsu, and *if* this tephra could be reliably linked to a specific eruption independently dated elsewhere, then the age for that eruption could

be used to 'reset' the accumulated errors below and above that tephra layer. A big and unrealised 'if', but this would certainly help reducing the errors.

Abstract line 10 (also p4 line 1,3,5), the Suigetsu dataset is not presented as a calibration curve. It is a dataset that can contribute to a $^{14}$C calibration curve (and indeed forms part of the IntCal13 calibration curve). On p8 you correctly name the Suigetsu time-series as a comparison curve – please use this term instead of calibration curve throughout the manuscript. I would also disagree with your suggestion to start calibrating radiocarbon ages with Suigetsu. The IntCal curve remains the internationally ratified curve and remains recommended for calibration of $^{14}$C dates. It has more dates than just the Suigetsu ones, and it contains a model to derive calendar age and radiocarbon age uncertainties.

p3 line 31&33, do you really mean $\delta^{14}$C here, or rather $^{14}$C? The measurements are in $^{14}$C, and $\delta^{14}$C involves estimating the calendar timing and from this calculate atmospheric $^{14}$C concentration at a series of points in time. Therefore, $\delta^{14}$C values depend on the record's age-model (i.e. a given $^{14}$C age will result in a different $\delta^{14}$C value at a different calendar age $t$). Please clarify this in your manuscript. Also, is this time-dependency properly accounted for in Fig. 4B surely your $\delta^{14}$C clouds should be sloping?

p2 line 20–22, also p7 line 1 and p10 lines 1–2, this is not entirely new. Previous literature has developed approaches to visualise dating uncertainty of proxy records, e.g. Blaauw et al. 2007 (doi:0.1177/0959683607075857, Fig. 2), even though they do not include the uncertainties of proxy measurements. For absolute and relative errors in ice cores see also Figs 4 and 5 of Blaauw et al. 2010 (doi:10.1002/jqs.1330).

Details

p2 line 3, not all layered records are varves (e.g., trees, ice)

p2 line 19, what are bifurcation parameters? Needs more explanation for Climate of

the Past readers, or alternatively left out

p5 line 14, what is a Riemann sum? Provide a reference or explain for Climate of the Past readers

Language p5 line 19, chosen
* * *

---

## Author Comment (AC2) · 31 Jul 2017

We thank the reviewer for her/his positive evaluation of our manuscript. Regarding your two main points:

1. We agree that a representation of the temporal evolution of uncertainties in terms of $\hat{x}$ is misleading, and our main point is that the sequence of probability density functions $p(x|t)$ themselves should in fact be the focus of representing this evolution. This point is already mentioned in the manuscript, but in the revised version, we will emphasize it even further.

2. The blue shadings in Fig. 2 indicate the probability densities $p(x|t)$ for each $t$, and not $p(t|z)$; this is also noted in the color bar label. The densities $p(x|t)$ are

the final results of our approach to represent the record as probability densities (over the proxy values $x$) for different, error-free ages $t$. The dating uncertainty distribution $p(t|z)$ is used in the derivation of $p(x|t)$ via Eq. (1). Furthermore, the increase of dating uncertainties towards the past, which is apparent in Fig. 1B, is reflected in the increasing spread of $p(x|t)$ the further one goes into the past in Figs. 2C and 2D. This result is not trivial: it is a consequence of the Bayesian approach we employ. We will clarify this point further in the revised version.

Regarding the minor points in the Review:

For each annual layer, it is indeed possible that the uncertainty distributions are skewed, just as Referee #1 points out. In the original study (Andersen et al., QSR, 2006) reporting the chronology employed herein, uncertain layers are counted as $1/2 \pm 1/2$, thereby assuming a symmetric distribution. This counting should be adjusted in cases where the probability to miss a layer and the probability to count a false layer are not identical. The maximum counting error would then generally not be the same for negative and positive values, and the overall uncertainty distribution $p(t|z)$, which would be accordingly skewed, should be used when computing $p(x|t)$.

We would like to emphasize here that any functional form for $p(t|z)$ can be used in our approach. We thank the reviewer for pointing this out, and will add a corresponding sentence in the revised manuscript.

P2L2: We will correct this in the revised manuscript.

P2L9: We agree with the reviewer that in our approach to represent dating uncertainties, abrupt transitions that actually exist in the proxy evolution will be typically smoothed out in accordance with the uncertain dating, rather than being artificially amplified. However, in traditional proxy record representations, proxy values

are shown at specific time points. Ignoring the uncertainties of these time points may lead to situations where transitions appear much sharper than is actually supported by the data themselves when the dating uncertainties are considered.

P5L16: The Riemann sum, which is used to approximate the integral in this discrete setting, is defined using $r_i = (z_{i+1} - z_{i-1})/2$. This is the average of the two increments above and below the depth $z_i$, which the reviewer refers to as $\Delta z_i$: Setting $\Delta z_i := z_{i+1} - z_i$, we have $r_i = (z_{i+1} - z_{i-1})/2 = ((z_{i+1} - z_i) + (z_i - z_{i-1}))/2 = (\Delta z_i + \Delta z_{i-1})/2$. Taking this average is the standard approach when approximating a (continuous) integral by a discrete sum. It provides a better approximation than taking only the previous (or the following) increment. We will clarify this in the revised version.

P5L19: Thank you, this will be corrected.

P7L10: Please refer to our author comment AC1. We had, due to a typo, in fact uploaded an erroneous version of Fig. 2. In the figure attached to AC1, which will also be used in the revised version, small-scale differences between Fig. 2C and Fig. 2D are clearly visible.

Figure 1A: We had also thought about a 3D figure for the sketch of our method, and we appreciate the reviewer's effort in suggesting one. Figure 1 attached to this comment would be a possible 3D version of the sketch. However, it is actually misleading to use 3D cartesian coordinates because the $z$-axis is integrated over, and does thus play a different role than the $x$- and $t$-axes. In order to avoid confusion, we would therefore like to keep the original 2D version.

Figure 1B: As the reviewer notes correctly, this change in slope occurs at the transition from glacial to interglacial conditions, and reflects the substantial increase in dating uncertainties at this point, due to changing accumulation rates and increasing pressure in the ice. We will add a corresponding sentence in the revision.

Figure 2: We thank the reviewer for pointing this out to us and will use identical y-axes in the revised version of our study.

If the editor agrees, we will revise our manuscript in accordance with the reviewer's comments and our responses above.
* * *
[Figure]

$$p(x|t) \propto \sum_{i=1}^{N} r_i p(x|z_i) p(t|z_i)$$

**Fig. 1.**

---

## Author Comment (AC4) · 31 Jul 2017

We thank the reviewer for the thorough evaluation of our manuscript. Regarding her/his specific comments:

We agree that, realistically, the probability of missing an annual layer is generally not identical to counting a false one. However, for both the NGRIP and the Suigetsu datasets, the reported dating uncertainties are based on the assumption of symmetric counting errors. It would be interesting to repeat the annual layer counting for these records, taking into account that the error distributions are most likely not symmetric. Non-symmetric errors would lead to different maximum counting errors for negative and positive values, and hence to the overall

(i.e., cumulative) dating uncertainty distributions p(t|z) being skewed. This skewness of p(t|z) could then be used when applying our method in just the same way, as pointed out by Referee #1. Ideally, the correct p(t|z) would be reported along-side with the record itself; our formalism can treat arbitrary dating uncertainty distributions, and its specific characteristics will be propagated to the final p(x|t). We will add a corresponding comment in the revised version of the manuscript.

- nes 15-16: We agree with the reviewer. A tephra layer would indeed reset the MCE back to zero, which would be reflected in very narrow dating uncertainty distributions p(t|z) around such a layer. The spread in the derived p(x|t) would decrease accordingly in Figs. 2C, 2D, 4B and 4D. We will add a sentence on this point in the revision.
- bs line 10: We apologize for the incorrect usage of "calibration curve," and will replace it by "comparison curve" throughout the revised version. We will also change our suggestion to actually use the Suigetsu record as a calibration curve, since we do agree that IntCal should be used for such a purpose. In the revised manuscript, we would still show how to use our Bayesian method to derive the overall uncertainties of the radiocarbon ages — including the errors from the layer-counting process — and propose that this could also be done for the IntCal calibration curve.
- ne 31&33: In fact, we use the notation  $\Delta^{14}$ C for the "inferred level of radiocarbon in the atmosphere," given in **per mil**, as deviations from the reference value of 1950, in the same way as in the associated reference (Bronk Ramsey et al., Science, 2012). We agree that, to infer these values, information about timing needs to be included. For the time interval under study, the Suigetsu chronology is derived from the varve-counting process, and this temporal information is used to infer past levels of atmospheric radiocarbon (i.e.  $\Delta^{14}$ C), as deviation from the 1950 reference value. Thus, this record consists of inferred  $\Delta^{14}$ C values, associated
with corresponding sediment depth values and ages, including age uncertainties. This is all the information we need in order to apply our Bayesian approach to derive p(x|t), as shown in Fig. 4B.

line 20-22: We agree with the reviewer that several studies have presented approaches to visualize dating uncertainties of paleoclimatic proxies. We already cited two studies by M. Blaauw (2010 & 1012), but will also include the 2007 reference in the revised version. In the revised manuscript, we will also note more explicitly that the idea of directly visualizing dating uncertainties is not new. However, we would like to emphasize that, to the best of our knowledge, visualization of proxy record uncertainties (in the style of Blaauw et al., 2007) is not widely used in the geoscientific community that deals with layer-counted proxy archives. With our proposed approach, we are not only able to visualise the proxy record uncertainties, but also to *quantify* them in a mathematically precise sense: p(x|t) yields the best estimation of x at time t, given the observed data and their uncertainties. Therefore, the derived p(x|t) series can also be used for further, quantitative analyses.

p2 line 3: The reviewer is right, we will correct this in the revision.

- p2 line 19: By "bifurcation parameters", we refer to the estimated parameters of an energybalance model in the corresponding reference. We will rephrase the sentence accordingly.
- p5 line 14: A Riemann sum is the standard technique for approximating a (continuous) integral, given discrete values of the function to be integrated. We will clarify this in the revised manuscript and provide a standard, first-year calculus reference.

p5 line 19: Thank you, this will be corrected in the revised version.

If the editor agrees, we will revise our manuscript in accordance with the reviewer's comments and our responses above.

CPD

---

## Author Response (AR1)

**Point-by-point responses to the referees' comments**

**Response to Anonymous Referee # 1:**

We thank the reviewer for her/his positive evaluation of our manuscript. Regarding your two main points:

1. We agree that a representation of the temporal evolution of uncertainties in terms of $\hat{x}$ is misleading, and our main point is that the sequence of probability density functions $p(x|t)$ themselves should in fact be the focus of representing this evolution. This point was already mentioned in the manuscript, but in the revised version, we have emphasized it even further.

2. The blue shadings in Fig. 2 indicate the probability densities $p(x|t)$ for each $t$, and not $p(t|z)$; this is also noted in the color bar label. The densities $p(x|t)$ are the final results of our approach to represent the record as probability densities (over the proxy values $x$) for different, error-free ages $t$. The dating uncertainty distribution $p(t|z)$ is used in the derivation of $p(x|t)$ via Eq. (1). Furthermore, the increase of dating uncertainties towards the past, which is apparent in Fig. 1B, is reflected in the increasing spread of $p(x|t)$ the further one goes into the past in Figs. 2C and 2D. This result is not trivial: it is a consequence of the Bayesian approach we employ. We have clarified this point further in the revised version.

Regarding the minor points in the Review:

For each annual layer, it is indeed possible that the uncertainty distributions are skewed, just as Referee #1 points out. In the original study (Andersen et al., QSR, 2006) reporting the chronology employed herein, uncertain layers are counted as $1/2 \pm 1/2$, thereby assuming a symmetric distribution. This counting should be adjusted in cases where the probability to miss a layer and the probability to count a false layer are not identical. The maximum counting error would then generally not be the same for negative and positive values, and the overall uncertainty distribution $p(t|z)$, which would be accordingly skewed, should be used when computing $p(x|t)$.

We would like to emphasize here that any functional form for $p(t|z)$ can be used in our approach. We thank the reviewer for pointing this out, and have added a corresponding sentence in the revised manuscript.

P2L2: We have corrected this in the revised manuscript.

P2L9: We agree with the reviewer that in our approach to represent dating uncertainties, abrupt transitions that actually exist in the proxy evolution will be typically smoothed out in accordance with the uncertain dating, rather than being artificially amplified. However, in traditional proxy record representations, proxy values are shown at specific time points. Ignoring the uncertainties of these time points may lead to situations where transitions appear much sharper than is actually supported by the data themselves when the dating uncertainties are considered.

P5L16: The Riemann sum, which is used to approximate the integral in this discrete setting, is defined using $r_i = (z_{i+1} - z_{i-1})/2$. This is the average of the two increments above and below the depth $z_i$, which the reviewer refers to as $\Delta z_i$: Setting $\Delta z_i := z_{i+1} - z_i$, we have $r_i = (z_{i+1} - z_{i-1})/2 = ((z_{i+1} - z_i) + (z_i - z_{i-1}))/2 = (\Delta z_i + \Delta z_{i-1})/2$. Taking this average is the standard approach when approximating a (continuous) integral by a discrete sum. It provides a better approximation than taking only the previous (or the following) increment. We have clarified this in the revised version.

P5L19: Thank you, this has been corrected.

P7L10: Please refer to our author comment AC1. We had, due to a typo, in fact uploaded an erroneous version of Fig. 2. In the figure attached to AC1, which is also used in the revised version, small-scale differences between Fig. 2C and Fig. 2D are clearly visible.

Figure 1A: We have, in the revised version of the manuscript, replaced the 2D figure by a 3D figure for the sketch of our method, and we appreciate the reviewer's effort in suggesting one. It might, however, be slightly misleading to use 3D cartesian coordinates because the $z$-axis is integrated over and thus plays a different role than the $x$- and $t$-axes. In order to avoid confusion, we have added a comment in the caption of the figure.

Figure 1B: As the reviewer notes correctly, this change in slope occurs at the transition from glacial to interglacial conditions, and reflects the substantial increase in dating uncertainties at this point, due to changing accumulation rates and increasing pressure in the ice. We have added a corresponding sentence in the revision.

Figure 2: We thank the reviewer for pointing this out to us and use identical y-axes in the revised version of our study.

**Response to Anonymous Referee # 2**

We thank the reviewer for the thorough evaluation of our manuscript. Regarding her/his specific comments:

> We agree that, realistically, the probability of missing an annual layer is generally not identical to counting a false one. However, for both the NGRIP and the Suigetsu datasets, the reported dating uncertainties are based on the assumption of symmetric counting errors. It would be interesting to repeat the annual layer counting for these records, taking into account that the error distributions are most likely not symmetric. Non-symmetric errors would lead to different maximum counting errors for negative and positive values, and hence to the overall (i.e., cumulative) dating uncertainty distributions $p(t|z)$ being skewed. This skewness of $p(t|z)$ could then be used when applying our method in just the same way, as pointed out by Referee #1. Ideally, the correct $p(t|z)$ would be reported alongside with the record itself; our formalism can treat arbitrary dating uncertainty distributions, and its specific characteristics will be propagated to the final $p(x|t)$. We have added a corresponding comment in the revised version of the manuscript.

p8 lines 15-16: We agree with the reviewer. A tephra layer would indeed reset the MCE back to zero, which would be reflected in very narrow dating uncertainty distributions $p(t|z)$ around such a layer. The spread in the derived $p(x|t)$ would decrease accordingly in Figs. 2C, 2D, 4B and 4D. We have added a sentence on this point in the revision.

Abs line 10: We apologize for the incorrect usage of "calibration curve," and will replace it by "comparison curve" throughout the revised version. We will also change our suggestion to actually use the Suigetsu record as a calibration curve, since we do agree that IntCal should be used for such a purpose. In the revised manuscript, we still show how to use our Bayesian method to derive the overall uncertainties of the radiocarbon ages — including the errors from the layer-counting process — and propose that this could also be done for the IntCal calibration curve.

p3 line 31&33: In fact, we use the notation $\Delta^{14}$C for the "inferred level of radiocarbon in the atmosphere," given in **per mil**, as deviations from the reference value of 1950, in the same way as in the associated reference (Bronk Ramsey et al., Science, 2012). We agree that, to infer these values, information about timing needs to be included. For the

time interval under study, the Suigetsu chronology is derived from the varve-counting process, and this temporal information is used to infer past levels of atmospheric radiocarbon (i.e. $\Delta^{14}$C), as deviation from the 1950 reference value. Thus, this record consists of inferred $\Delta^{14}$C values, associated with corresponding sediment depth values and ages, including age uncertainties. This is all the information we need in order to apply our Bayesian approach to derive $p(x|t)$, as shown in Fig. 4B.

p2 line 20-22: We agree with the reviewer that several studies have presented approaches to visualize dating uncertainties of paleoclimatic proxies. We already cited two studies by M. Blaauw (2010 & 1012), but will also include the 2007 reference in the revised version. In the revised manuscript, we also note more explicitly that the idea of directly visualizing dating uncertainties is not new. However, we would like to emphasize that, to the best of our knowledge, visualization of proxy record uncertainties (in the style of Blaauw et al., 2007) is not widely used in the geoscientific community that deals with layer-counted proxy archives. With our proposed approach, we are not only able to visualise the proxy record uncertainties, but also to *quantify* them in a mathematically precise sense: $p(x|t)$ yields the best estimation of $x$ at time $t$, given the observed data and their uncertainties. Therefore, the derived $p(x|t)$ series can also be used for further, quantitative analyses.

p2 line 3: The reviewer is right, we have corrected this in the revision.

p2 line 19: By "bifurcation parameters", we refer to the estimated parameters of an energy-balance model in the corresponding reference. We have rephrased the sentence accordingly.

p5 line 14: A Riemann sum is the standard technique for approximating a (continuous) integral, given discrete values of the function to be integrated. We have clarified this in the revised manuscript and provide a standard, first-year calculus reference.

p5 line 19: Thank you, this has been corrected in the revised version.

[revised manuscript text omitted]